# Cyclin-Dependent Kinase and Antioxidant Gene Expression in Cancers with Poor Therapeutic Response

**DOI:** 10.3390/ph13020026

**Published:** 2020-02-05

**Authors:** George S. Scaria, Betsy T. Kren, Mark A. Klein

**Affiliations:** 1Research Service, Minneapolis VA Health Care System, Minneapolis, MN 55417, USA; scari005@umn.edu (G.S.S.); krenx001@umn.edu (B.T.K.); 2Division of Hematology, Oncology and Transplantation, Department of Medicine, University of Minnesota, Minneapolis, MN 55455, USA; 3Hematology/Oncology Section, Primary Care Service Line, Minneapolis VA Health Care System, Minneapolis, MN 55417, USA

**Keywords:** hepatocellular carcinoma, pancreatic neoplasms, mesothelioma, genome, cell cycle, reactive oxygen species, mitochondria

## Abstract

Pancreatic cancer, hepatocellular carcinoma (HCC), and mesothelioma are treatment-refractory cancers, and patients afflicted with these cancers generally have a very poor prognosis. The genomics of these tumors were analyzed as part of The Cancer Genome Atlas (TCGA) project. However, these analyses are an overview and may miss pathway interactions that could be exploited for therapeutic targeting. In this study, the TCGA Pan-Cancer datasets were queried via cBioPortal for correlations among mRNA expression of key genes in the cell cycle and mitochondrial (mt) antioxidant defense pathways. Here we describe these correlations. The results support further evaluation to develop combination treatment strategies that target these two critical pathways in pancreatic cancer, hepatocellular carcinoma, and mesothelioma.

## 1. Introduction

Mesothelioma, hepatocellular carcinoma (HCC), and pancreatic cancers are three of the most devastating cancers; these diseases are also recalcitrant to treatment [1,2,3]. The median survival for patients afflicted with these cancers at advanced stages range primarily from months to approximately one year [1,2,3]. In settings where local, potentially curative options are not available (a significant proportion of cases of mesothelioma, hepatocellular, and pancreatic cancers), systemic therapy with a targeted agent or chemotherapy are the standard treatment options [4,5,6]. For mesothelioma, the standard first-line therapy is the chemotherapy regimen cisplatin (involved in DNA adduct formation) plus the antifolate agent pemetrexed (with or without the angiogenesis inhibitor bevacizumab) [3,7]. For hepatocellular carcinoma, first-line options include the tyrosine kinase inhibitors (TKIs) sorafenib or lenvatinib [8]. In pancreatic cancer, the chemotherapy regimen called FOLFIRINOX (5-fluorouracil, oxaliplatin, and irinotecan) is associated with the highest overall survival in patients with advanced disease [9]. Years of clinical trials have not yielded significant advances, and to date, immunotherapy has had modest benefit in mesothelioma and HCC [10,11]. Thus, new approaches against these cancers are needed. Two cellular pathways that are promising as targets in these cancers include (1) the cell cycle pathway and related genes/proteins and (2) mitochondrial (mt) antioxidant defense [12,13]. 

### 1.1. Cell Cycle Regulation in Cancer 

Among the various players active in the cell cycle [12,14], cyclin D1 and CDK4/6 (cyclin-dependent kinases 4 and 6) are major proteins responsible for progression through G1 to S phases, and regulation of this step is corrupted in many cancers [13,15]. Cyclin D1 binds to CDK4 or CDK6 and these complexes promote phosphorylation of retinoblastoma protein (Rb). Additional cyclin–CDK complexes (such as CDK2/cyclin E1) further phosphorylate Rb, which allows transcription factors to become active and thereby drive cell cycle progression. The CDK inhibitor p16INK4a (p16, a protein) inhibits the cyclin D1–CDK4 or cyclin D1–CDK6 complexes [14,15]. Inactivation of p16 appears to promote the pathogenesis of many tumor types, including mesothelioma, breast cancer, pancreatic cancer, non-small cell lung cancer, esophageal cancer, and head and neck cancer [14,15,16]. Several studies confirm that p16 loss is extremely common in mesothelioma [17]. Deletion of the 9p21 locus that encodes p16 was deleted in 35/40 cases (88%) in one study [17]. Overexpression of the CDK4/6 partner cyclin D1 has been identified in a number of tumor types [14,15,18], such as mantle cell lymphoma (with a well-known translocation involving cyclin D1 in nearly 100% of these cases), non-small cell lung cancer, and breast cancer. 

### 1.2. Mitochondrial Antioxidant Defense

Thioredoxin 2 (Trx2) plays an essential role in mitochondrial (mt) and cell viability, and an essential role for Trx2 in the response to oxidative stress is well supported in the literature [19,20]. Trx2 haploinsufficient (Trx2 +/−) mice show impaired mt function, increased mt oxidative stress, decreased ATP production, and increased oxidative damage to nuclear DNA, lipids, and proteins [21]. TNF-α-induced reactive oxygen species (ROS) generation, NF-κB activation [22], mitochondrial permeability transition (mPT) [23], and apoptosis [24] can all be regulated by Trx2. Finally, overexpression of Trx2 enhances mt membrane potential (∆ψm) [25]. Auranofin, a systemic therapeutic molecule that was developed for rheumatoid arthritis, represses disease progression via decreased inflammation and increased cell-mediated immunity. Its main mechanism of action is the inhibition of the reduction of Trx2 by thioredoxin reductase 2 [26,27,28], thereby defeating the ability of maintaining low intracellular reactive oxygen species (a key adaptation for cancer cell survival). HCC develops in the context of chronic inflammatory liver disease and progression is characterized by an increasing immunosuppressive tumor environment, thereby implicating mitochondrial antioxidant defense as a viable target [29]. Furthermore, the abnormal vascularization of solid tumors results in the development of metabolically compromised microenvironments that severely limits the ability of the cancer cells to survive a decrease in mitochondrial function, suggesting that targeting mitochondrial antioxidant defense is a key component for eradicating the quiescent tumor cell population [30,31].

### 1.3. Interrelationship of Cell Cycle-Related Genes and Mitochondrial Antioxidant Defense Genes 

Novel approaches to combination therapies are needed due to the modest benefits of current treatments for mesothelioma, HCC, and pancreatic cancer as described previously. It can be challenging to design combination therapies for clinical studies based on the exorbitant number of potentially targetable pathways that may interact. We chose to focus on (1) pathways (or, more precisely, pathway-related genes that correspond to a potential drug target) that have an FDA-approved drug (such as palbociclib or auranofin), (2) at least one target of a two-target pair having a known association with the cancer (such as CDK4 in pancreatic cancer, mesothelioma, and HCC), (3) the fact that there is a plausible interaction between genes or gene products, and (4) novel combinations. Based on this, we chose to focus on cell cycle-related genes and mitochondrial antioxidant defense genes. Palbociclib and auranofin are both FDA-approved. CDK4 has been evaluated extensively in mesothelioma, HCC, and pancreatic cancer [32]. Palbociclib (PD-0332991), a selective CDK4/6 inhibitor, restricts tumor growth in preclinical models of HCC and pancreatic cancer [32,33]. Furthermore, the combined inhibition of CDK4/6 and PI3K/AKT/mTOR pathways inhibits mesothelioma cell growth [34]. The gene FOXM1 is known to be highly interactive with CDK2, CDK4, and thioredoxins [35,36]. For example, FOXM1 induces transcription of cyclin D1 and CDK4 to enhance activity of these proteins [37]. In addition, auranofin is a thiol compound that inhibits the thioredoxin pathway, at least partially via FOXM1 downregulation [38].

### 1.4. The Cancer Genome Atlas (TCGA)

The Cancer Genome Atlas has been a significant and widely utilized resource [39]. Many cancers, including mesothelioma, hepatocellular carcinoma, and pancreatic cancer, have been included in the TCGA efforts [40,41,42]. A web-based tool, cBioPortal, has been created and is continually updated to aid in the analysis of TCGA data [43,44]. Currently, experimental treatments do not take full advantage of the knowledge about the transcriptomics of these diseases. We interrogated the TCGA data using cBioPortal and employing a set of genes (Table 1) encompassing these two pathways to determine their potential utility in meeting the urgent need for new precision oncology-based treatment approaches for these diseases. 

## 2. Results

### 2.1. mRNA Expression 

Heatmaps of the genes were generated for all three cancers (Figure 1) with the z-score set to a threshold of 2.0 employing unsupervised hierarchical clustering and using the cBioPortal tool. Most notable was the clear distinction of cell cycle-related mRNA clustering in all three cancers.

Second, key non-cell cycle CDKs (CDK5 and CDK11B) clustered with key antioxidant genes, including thioredoxin 2 (TXN2), methionine sulfoxide reductase (MSRA), and thioredoxin reductase 2 (TXNRD2). mRNA expression was also compared utilizing Spearman and Pearson coefficients (two distinct, and complementary, correlation formulas available in cBioPortal) among key pathway genes for all three cancers to determine the strength of association (Appendix A). For mesothelioma, key interactions (either positive or negative, with Spearman and/or Pearson coefficients with absolute values > 0.3) included CDK2-TXN2 (neg), CDK2-FOXM1 (pos), CDK2-CSNK2A1 (gene for CK2) (pos), CDK4-NFKB1 (neg), CDK4-FOXM1 (pos), CDK5-TXN2 (pos), CDK5-GLRX2 (pos), CDK5-NFKB1 (neg), CDK5-CSNK2B (pos), TXN-GLRX2 (pos), TXN-CSNK2B (pos), TXN2-GLRX2 (pos), TXN2-CSNK2B (pos), and HIF1-TXN2 (neg). (GLRX2 encodes glutaredoxin 2, CSNK2A1 encodes the protein CK2A, CSNK2B encodes the protein CK2B, and HIF1 encodes hypoxia inducible factor 1.) For pancreatic cancer, notable associations included CDK2-TXN2 (neg), CDK2-FOXM1 (pos), CDK2-HIF1 (pos), CDK4-CSNK2B (pos), CDK5-TXN2 (pos), CDK5-TXNRD2 (pos), CDK5-GLRX2 (pos), CDK5-NFKB1 (neg), CDK5-CSNK2B (pos), TXN-GLRX2 (pos), TXN-CSNK2B (pos), TXN-GPX2 (pos), TXN-FOXM1 (pos), TXN2-TXNRD2 (pos), TXN2-CSNK2B (pos), HIF-TXN (neg), HIF-TXNRD1 (pos), HIF-TXN2 (neg), HIF-TXNRD2 (neg), HIF-SOD2 (pos), and HIF-NFKB1 (pos). (GPX2 encodes glutaperoxidase 2, TXN encodes thioredoxin 1, and SOD encodes superoxide dismutase.) For hepatocellular carcinoma, associations included CDK2-TXN2 (neg), CDK2-TXNRD2 (neg), CDK2-FOXM1 (pos), CDK2-CSNK2A1 (gene for CK2) (pos), CDK2-HIF1 (pos), CDK4-PRDX3 (neg), CDK4-CSNK2A1 (pos), CDK4-CSNK2B (pos), CDK5-GLRX2 (pos), CDK5-NFKB1 (neg), CDK5-CSNK2B (pos), CDK5-HIF1 (neg), TXN-TXNRD1 (pos), TXN-GLRX2 (pos), TXN-GPX2 (pos), TXN-SOD2 (pos), TXN2-NFKB1 (neg), TXN-CSNK2B (pos), TXN2-TXNRD2 (pos), TXN2-NFKB1 (neg), TXN2-CSNK2A1 (neg), TXN2-CSNK2B (pos), HIF1-TXN (neg), HIF1-TXN2 (neg), HIF1-TXNRD2 (neg), HIF1-NFKB1 (pos), HIF1-CSNK2A1 (pos), and HIF1-CSNK2B (neg). Please see the Appendix A for a tabular format of the above information.

### 2.2. Overall Survival

The Kaplan–Meier curves for overall survival of patients with each cancer with mRNA panel expression for z-scores > 2 vs. tumors with z-scores < 2 are shown in Figure 2A. Z-scores were applied to Kaplan–Meier curves based on the gene set from Table 1. Overall survival was significantly higher for patients with tumors exhibiting z-scores < 2 in hepatocellular carcinoma and mesothelioma, but overall survival was not higher for patients with pancreatic cancer. The survival curves encompass patients with all stages of the particular cancer evaluated.

### 2.3. Progression-free Survival

The Kaplan–Meier curves for progression-free survival of patients with cancers expressing the target mRNA with z-scores > 2 vs. tumors with z-scores < 2 are shown in Figure 2B. In contrast to the overall survival curves, alteration of mRNA expression of the selected gene set was not associated with progression-free survival for any of the cancers.

### 2.4. Copy Number Alterations

The top-ranked copy number variations (CNVs) were evaluated for all three cancers (Figure 3A). Unexpectedly, the top 3 copy number variants for both mesothelioma and pancreatic cancer were the same: CDKN2A (encoding p16INK4a), CDKN2B (encoding p15INK4b), MTAP (encoding methylthioadenosine phosphorylase), DMRTA1 (doublesex and mab-3 related transcription factor 1), and LINC01239 (long intergenic non-protein coding RNA 1239). In addition, three interferons (IFNA1, IFNA2, and IFNE) and MIR31HG (microRNA-31 Host Gene) were also in the top 10 for these two cancers. The interferon alpha gene alterations observed are likely because they cluster near band 9q21 that is proximal to the locus of CDKN2A, which was included in the screened gene set and is commonly deleted in both cancer sets. However, it is unclear whether the alteration of the IFN-alpha locus has any effect on tumor behavior. No overlap in top ranked CNVs between mesothelioma, pancreatic, and hepatocellular cancers were noted (Figure 3A).

### 2.5. Mutations

Co-expression analysis of mutations in the cancer datasets was performed as well. The top genes with mutations are shown in Figure 3B. The top 3 genes with mutations in pancreatic cancer included RAS and TP53, as expected, in addition to SMAD4 (SMAD family member 4). For mesothelioma, the top genes with mutations were BAP1 (BRCA-associated protein 1), NF2 (neurofibromin 2), and SETD2 (SET domain containing 2, histone lysine methyltransferase). Of note, TP53 was the 4th most common mutation in mesothelioma. For hepatocellular carcinoma, the top gene mutations were TP53, CTNNB1 (catenin beta 1), and TTN (titin). TTN was also the 4th most commonly mutated gene in pancreatic cancer. The top mutation profile for all three cancers reflected the two or three most common mutational drivers observed for each type of cancer as expected, with TP53 consistently playing a role in all of the cancer types [41,45,46]. Intriguingly, mitochondrial metabolism and ROS are considered essential for KRAS-mediated tumorigenicity [47].

## 3. Discussion

Mesothelioma, pancreatic cancer, and hepatocellular carcinoma are three of the most treatment-refractory cancers [1,2,3]. In the metastatic stages of each, there are no curative options, and current treatment options have very modest benefit [1,2,3]. New treatment paradigms are desperately needed. While TCGA analyses have been conducted previously for these tumors [40,41,42], descriptions of the TCGA-based analysis of each tumor cannot include all pathways of potential therapeutic application [40,41,42]. Based on previous studies, we chose to focus on select pathways that have therapeutic agents (the cell cycle and mitochondrial antioxidant defense), have not been analyzed in combination, and have not been studied thoroughly in treatment-refractory cancers. TCGA pairwise analysis with key pathway proteins indicated a notable correlation between the cell cycle CDKs 2 and 4 (negative) and CDK5 (positive) with mitochondrial antioxidant proteins in all three cancer types. Furthermore, NFKB1 and HIF1 expression were negatively correlated with CDK5 and TXN2 in all three cancers, while HIF1-NFKB1 showed a positive association with each other. Mounting evidence suggests that mitochondrial antioxidant defense plays a key role in the survival of non-replicating tumor initiating cells, intrinsic or acquired resistance to chemotherapeutic agents, and the metastases of tumors [48,49,50,51,52,53]. Furthermore, cell cycle CDKs 2 and 4 partner plus cyclins E and D1 have been shown to be downregulated in response to increased mitochondrial ROS or decreased ATP levels, both situations associated with TXN2 downregulation [54,55]. CDK4/cyclin D1 [56] and CDK4/cyclin E [57] are known to translocate to the mitochondria and can increase dramatically the activity of SOD2 (superoxide dismutase 2, Figure 4) in the absence of altering transcript or protein expression. It has been proposed that cell cycle progression is regulated in part via a mitochondria-mediated ROS mechanism [58]. Overall, cell cycle-related proteins and mitochondrial antioxidant defense proteins are important to the pathogenesis and growth of each of these cancers; therefore, we hypothesized that interactions may exist between these two pathways that may be amenable to targeting with agents already approved for other uses by the FDA. Drug “repurposing” (i.e., identification of new therapeutic applications for already approved drugs) is significantly more affordable and achievable, removing most of the expense, time, and high failure rate associated with novel drug development [59,60]. This strategy has great appeal for these three cancers due to their poor prognosis and limited market size, potentially achieving a rapid move to the clinic for identified new therapies.

## 4. Conclusions

Our study is the first to utilize the TCGA to evaluate the potential relationship between the cell cycle (and cell cycle-related proteins/genes) and mitochondrial antioxidant defense in three separate, treatment-refractory, cancers, namely pancreatic cancer, hepatocellular carcinoma, and mesothelioma. The potential relationships between the two pathways, based on key mRNA expression correlations and copy number variations, suggest that further laboratory-based studies to evaluate for synergistic interactions among inhibitors of each pathway may lead to new treatment strategies for at least a portion of these tumors. Of note, key inhibitors of the cell cycle pathway (i.e., palbociclib [16]) and mitochondrial antioxidant defense (i.e., auranofin [28]) are clinically approved by the FDA for other diseases; therefore, positive study results could lead to the design of early-stage clinical trials for these treatment-refractory cancers. 

## 5. Materials and Methods

All genomic data were obtained from The Cancer Genome Atlas (TCGA) via the web tool, cBioPortal [43,44]. A list of relevant genes from the cell cycle and mitochondrial antioxidant pathways was compiled by literature review and consensus of the authors [13,14,15,17,18,20,21,22,24,25,41,42,44]. The gene list was entered into the cBioPortal online tool. The online tool was used to identify overall survival and progression-free survival for the entire list of genes. Comparison is between genes with abnormalities (mutations, copy number variations, and mRNA expression levels). Subsequently, mRNA expression levels within tumors were compared with relevant mRNA expression levels of genes postulated to be related. Comparisons were made by Pearson’s and Spearman’s coefficients (tools included in cBioPortal) as it is unclear whether the postulated relationships are non-parametric or parametric. Additionally, the top copy number variations and mutations for each tumor type were also catalogued. 

## Figures and Tables

**Figure 1 pharmaceuticals-13-00026-f001:**
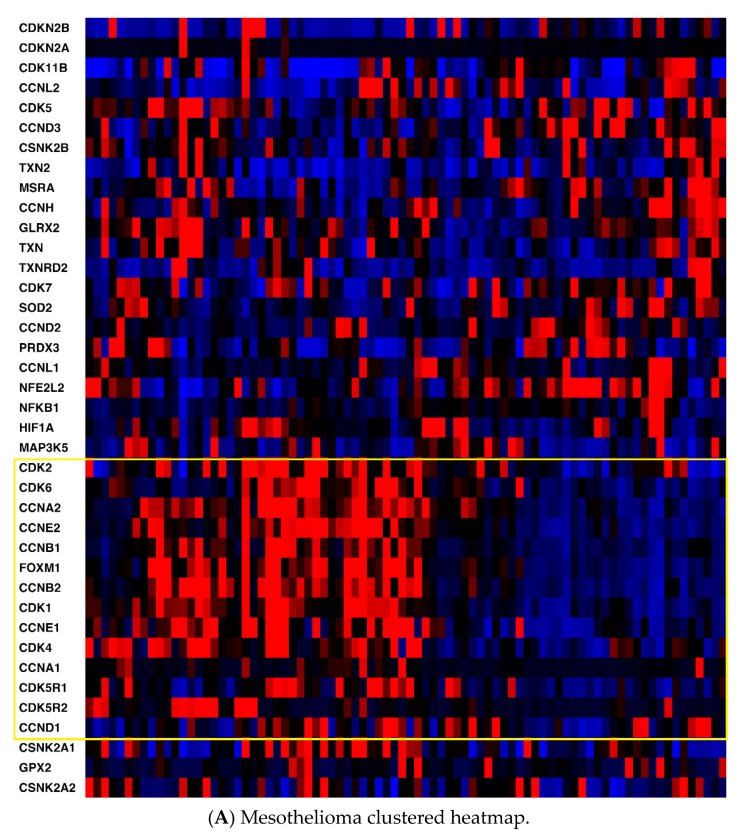
(**A**) Mesothelioma heatmap with cell cycle-related genes boxed; (**B**) Pancreatic cancer heatmap (cell cycle-related genes in the top box, antioxidant defense-related genes in the bottom box; (**C**) Hepatocellular carcinoma heatmap (cell cycle-related genes in the top box, antioxidant defense-related genes in the bottom box). Red, higher expression; blue, lower expression. Box is in yellow.

**Figure 2 pharmaceuticals-13-00026-f002:**
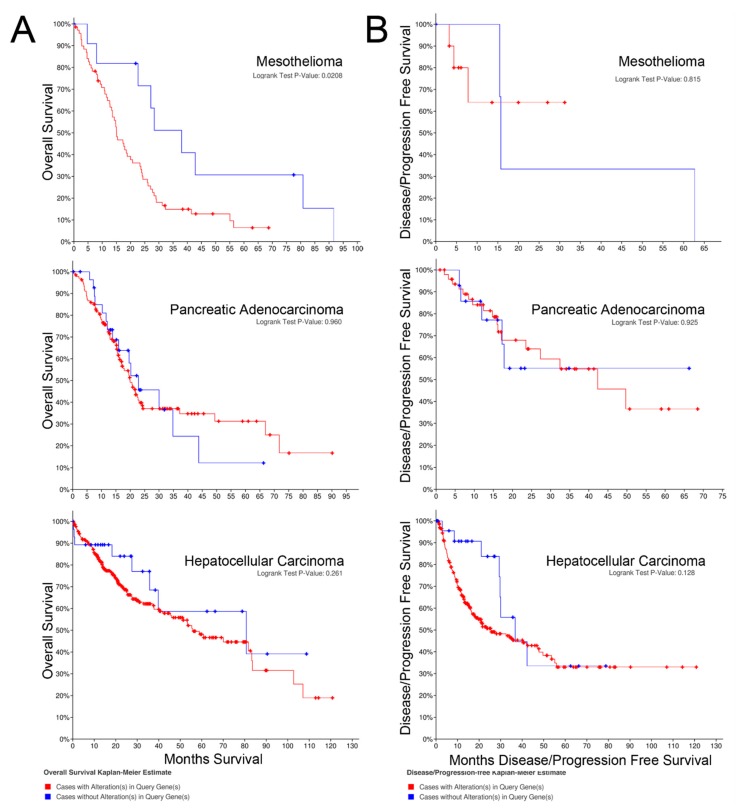
(**A**) Overall survival; (**B**) Progression-free survival.

**Figure 3 pharmaceuticals-13-00026-f003:**
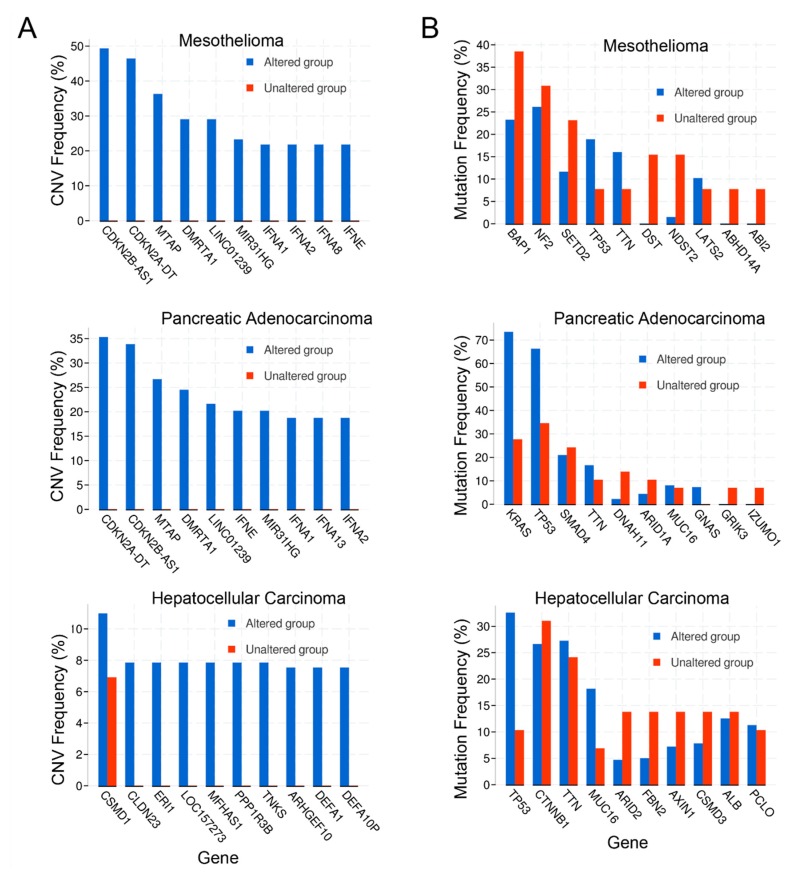
(**A**) Copy number variation (CNV) frequency; (**B**) Mutation frequency.

**Figure 4 pharmaceuticals-13-00026-f004:**
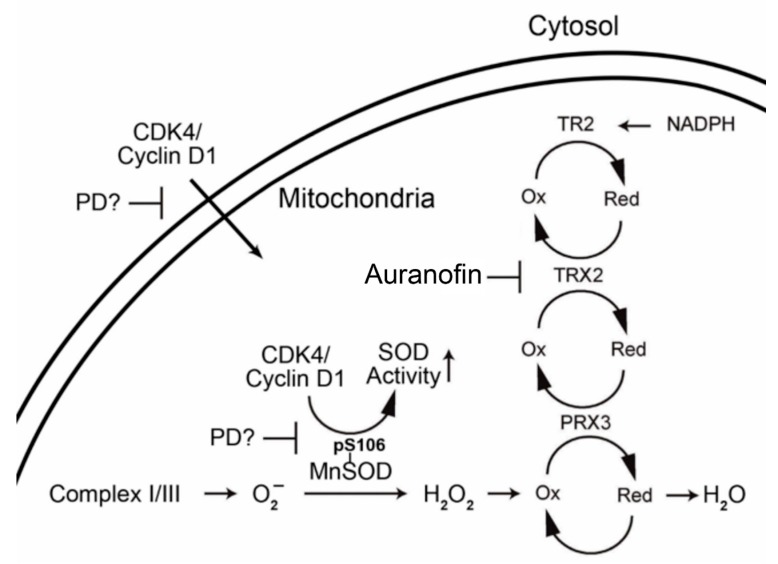
Proposed interaction of CDK4/cyclin D1 and mitochondrial antioxidant proteins.

**Table 1 pharmaceuticals-13-00026-t001:** Gene set employed for The Cancer Genome Atlas (TCGA) analysis.

CDKN2A	CDKN2B	CDK1	**CDK2**	**CDK4**	**CDK5**	CDK6	CDK7
CDK11B	CCNA1	CCNA2	CCNB1	CCNB2	CCND1	CCND2	CCND3
CCNE1	CCNE2	CDK5R1	CDK5R2	CCNH	CCNL1	CCNL2	**FOXM1**
**TXN**	**TXNRD1**	**TXN2**	**TXNRD2**	**PRDX3**	**GLRX2**	**GPX2**	**SOD2**
**MSRA**	**HIF1A**	**NFKB1**	**CSNK2A1**	**CSNK2A2**	**CSNK2B**	NFE2L2	MAP3K5

Genes indicated in bold type were used for the mRNA expression key pathway analysis.

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
