# Peer review of "Cyclin-Dependent Kinase and Antioxidant Gene Expression in Cancers with Poor Therapeutic Response"

_pharmaceuticals, 2020, doi:10.3390/ph13020026_

Round 1

Reviewer 1 Report

The article ‘’Cyclin-dependent Kinase and Anti-oxidant Gene Expression in Cancers with Poor Therapeutic Response’’ is a well-written article that highlights the role of CDKs and anti-oxidant gene expression in cancer with the poor therapeutic response of pancreatic, hepatocellular and mesothelioma cancer patients. Moreover, the authors have queried the TCGA Pan-Cancer datasets to correlate the mRNA expression of important genes in cell cycle and antioxidant mechanisms which is interesting. I would highly recommend this article for publication but after a minor revision which is highlighted below.

The authors should highlight the rationale for choosing particular genes for mRNA expression analysis in key pathways. Do other genes in the pathway were not included because they are not modulated? If not modulated it should be mentioned. The authors should provide the background on the therapies used for the treatment of pancreatic, hepatocellular and mesothelioma cancer patients. This would be necessary to understand because the mRNA expression changes, Copy number alterations are cancer type-dependent or treatment-induced response changes.

Author Response

We appreciate the reviewer's comments. In response, we have made the recommended changes by adding an additional section to the introduction to describe why we chose the two pathways. 

Reviewer 2 Report

The authors interrogated the TCGA Pan-Cancer datasets of mesothelioma, hepatocellular carcinoma, and pancreatic cancer, to seek a relationship between mRNA expression and CNAs of a panel of genes and patients’ outcome. The authors hypothesize that cell cycle pathways can interact with the mitochondrial antioxidant function, with potential implications for novel clinical strategies.

This is a very descriptive work that, although it shall requiring further experimental investigation, is worthy of publication.

This reviewer’s unique suggestion is to evidence the groups of gene expression clustering, which do not appear much clear in Figure 1.

Author Response

We appreciate the reviewer's comments. To clarify the heatmaps, we created yellow boxes to denote the relevant gene clusters. 

Reviewer 3 Report

In this manuscript, Scaria et al. using TCGA data to identify genes of certain pathways/classes that represent novel interactions in three treatment refractory cancers. Whilst the data shows some potentially interesting gene combinations, as stated by the authors this is merely correlative and thus I struggle to see the impact of the work on the field. Additionally, I do not believe the reason for choosing cell cycle related genes and mitochondrial antioxidant defence genes is convincingly explained.

As many different signalling networks drive the cell cycle, 'cell cycle pathway' is a misleading term. Cell cycle related genes is more suitable. Line 49 - gene names should be described upon first use. For the heat maps, the genes should be in the same order for each cancer so that the clustering can easily be seen. The description of Spearman and Pearson analysis is very unclear and cluttered - would be much better to organise the data in a table. Line 118 - top 5 genes, not 3 as stated (5 genes are listed)

Author Response

We appreciate the reviewer's comments. To clarify the choice of pathways included in the analyses, we added a section to the introduction for further explanation. For genes, we added the gene names to further clarify. For Spearman and Pearson's coefficients, those tools are part of cBioPortal, and we described that. To clarify the heat maps, we added a yellow box to single out each relevant cluster. Since the genes are clustered, we felt that changing the order would be a bit confusing. Also, we made a supplemental data table and submitted that with the original submission to clarify key potential interactions. 

Round 2

Reviewer 3 Report

The authors have made efforts to incorporate my suggested changes.